# Emulator-based Bayesian optimization for efficient multi-objective calibration of an individual-based model of malaria

Theresa Reiker[1,2], Monica Golumbeanu[1,2], Andrew Shattock[1,2], Lydia Burgert[1,2], Thomas A. Smith [1,2], Sarah Filippi[3], Ewan Cameron[4,5,6] & Melissa A. Penny [1,2✉]

Individual-based models have become important tools in the global battle against infectious diseases, yet model complexity can make calibration to biological and epidemiological data challenging. We propose using a Bayesian optimization framework employing Gaussian process or machine learning emulator functions to calibrate a complex malaria transmission simulator. We demonstrate our approach by optimizing over a high-dimensional parameter space with respect to a portfolio of multiple fitting objectives built from datasets capturing the natural history of malaria transmission and disease progression. Our approach quickly outperforms previous calibrations, yielding an improved final goodness of fit. Per-objective parameter importance and sensitivity diagnostics provided by our approach offer epidemiological insights and enhance trust in predictions through greater interpretability.

[1] Swiss Tropical and Public Health Institute, Basel, Switzerland. [2] University of Basel, Basel, Switzerland. [3] Imperial College London, London, UK. [4] Malaria Atlas Project, Big Data Institute, University of Oxford, Oxford, UK. [5] Curtin University, Perth, Australia. [6] Telethon Kids Institute, Perth Children's Hospital, Perth, Australia. ✉email: melissa.penny@unibas.ch

Over the last century, mathematical modeling has become an important tool to analyze and understand disease-dynamics and intervention-dynamics for many infectious diseases. Individual-based models (IBMs), where each person is simulated as an autonomous agent, are now widely used. These mathematical models capture heterogeneous characteristics and behaviors of individuals, and are often stochastic in nature. This bottom-up approach of simulating individuals and transmission events enables detailed, robust, and realistic predictions on population epidemic trajectories as well as the impact of interventions such as vaccines or new drugs[1,2]. Going beyond simpler (compartmental) models to capture stochasticity and heterogeneity in populations, disease progression, and transmission, IBMs can additionally account for contact networks, individual care seeking behavior, immunity effects, or within-human dynamics[1–3]. As such, well-developed IBMs provide opportunities for experimentation under relatively naturalistic conditions without expensive clinical or population studies. Prominent recent examples of the use of IBMs include assessing the benefit of travel restrictions during the Ebola outbreak 2014–2016[4] and guiding the public health response to the Covid-19 pandemic in multiple countries[5]. IBMs have also been applied to tuberculosis[6], influenza[7], dengue[8], and many other infectious diseases[2]. Within the field of malaria, several IBMs have been developed over the last 15 years and have been used to support understanding disease and mosquito dynamics[9–11], predict the public health impact or carry out economic analyses of (new) interventions[12–15]; and investigate drug resistance[16]. Many have had wide-reaching impact, influencing WHO policy recommendations[12,17–19] or strategies of national malaria control programs[20].

For model predictions to be meaningful, modelers need to ensure their models accurately capture abstractions of the real world. The potential complexity and realism of IBMs often come at the cost of long simulation times and potentially large numbers of input parameters, whose exact values are often unknown. Parameters may be unknown because they represent derived mathematical quantities that cannot be directly measured or require elaborate, costly experiments (for example shape parameters in decay functions[21]), because the data required to derive them in isolation is incomplete or accompanied by inherent biases, or because they interact with other parameters.

Calibrating IBMs poses a complex high-dimensional optimization problem and thus algorithm-based calibration is required to find a parameter set that ensures realistic model behavior, capturing the biological and epidemiological relationships of interest. Local optima may exist in the potentially highly irregular, high-dimensional goodness-of-fit surface, making iterative, purely sampling-based algorithms (e.g., Particle Swarm Optimization or extensions of Newton–Raphson) inefficient and, in light of finite runtimes and computational resources, unlikely to find global optima. Additionally, the *curse of dimensionality* means the number of evaluations of the model scales exponentially with the number of dimensions[22]. As an example, for the model discussed in this paper, a 23-dimensional parameter space at a sampling resolution of one sample per 10 percentile cell in each dimension, would yield $10^{\text{number of dimensions}} = 10^{23}$ cells. This is larger than number of stars in the observable Universe (of order $10^{22}$ [23]). Furthermore, most calibrations are not towards one objective or dataset. For multi-objective fitting, each parameter set requires the evaluation of multiple outputs and thus multiple simulations to ensure that all outcomes of interest are captured (in the model discussed here epidemiological outcomes such as prevalence, incidence, or mortality patterns).

In this study, we applied our approach to calibrate a well-established and used IBM of malaria dynamics called *Open-Malaria*. Malaria IBMs in particular are often highly complex (e.g., containing multiple sub-modules and many parameters), consider a two-host system influenced by seasonal dynamics, and often account for multifaceted within-host dynamics. OpenMalaria features within-host parasite dynamics, the progression of clinical disease, development of immunity, individual care seeking behavior, vector dynamics and pharmaceutical and non-pharmaceutical antimalarial interventions at vector and human level (https://github.com/SwissTPH/openmalaria/wiki)[3,21,24]. Previously, the model was calibrated using an asynchronous genetic algorithm (GA) to fit 23 parameters to 11 objectives representing different epidemiological outcomes, including age-specific prevalence and incidence patterns, age-specific mortality rates and hospitalization rates[3,21,24] (see Supplementary Notes 1 and 2 for details on the calibration objectives and data). However, the sampling-based nature and sequential function evaluations of GAs can be too slow for high-dimensional problems in irregular spaces where only a limited number of function evaluations are possible and valleys of neutral or lower fitness may be difficult to cross[25,26].

Other solutions to fit similarly detailed IBMs of malaria employ a combination of directly extracting parameter values from the literature where information is available, and fitting the remainder using multi-stage, modular Bayesian Markov Chain Monte Carlo (MCMC)-based methods[27–32]. For these models, multiple fitting objectives are often not addressed simultaneously. Rather, to our knowledge, most other malaria IBMs are divided into functional modules (such as the human transmissibility model, within-host parasite dynamics model, and the mosquito or vector model), which are assumed to be influenced by only a limited number of parameters each. The modules are then fit independently and in a sequential manner[28–32]. Modular approaches reduce the dimensionality of the problem, allowing for the use of relatively straightforward MCMC algorithms. However, these approaches struggle with efficiency in high dimensions as their Markovian nature requires many sequential function evaluations ($10^4$–$10^7$ even for simple models), driving up computing time and computational requirements[33]. Additionally, whilst allowing for the generation of posterior probability distributions of the parameters[31], the modular nature makes sequential approaches generally unable to account for interdependencies between parameters assigned to different modules and how their co-variation may affect disease dynamics.

Progress in recent years on numerical methods for supervised, regularized learning of smooth functions from discrete training data allows us to revisit calibration of detailed mathematical models using Bayesian methods for global optimization[34]. Current state-of-the art calibration approaches for stochastic simulators are often based around Kennedy and O'Hagan's (KOH) approach[35], where a posterior distribution for the calibration parameters is derived through a two-layer Bayesian approach involving cascade of surrogates (usually Gaussian processes, GPs)[36]. A first GP is used to model the systematic deviation between the simulator and the real process it represents, while a second GP is used to emulate the simulator[37]. However, this approach is computationally intense when scaling to high-dimensional input spaces and multi-objective optimization. A fully Bayesian KOH approach is likely computationally heavy[37] for the efficient calibration of detailed malaria simulators like OpenMalaria. Single-layer Bayesian optimization with GPs on the other hand have gained popularity as an efficient approach to tackle expensive optimization problems, for example in hyperparameter search problems in machine learning[38,39]. Assuming that the parameter-solution space exhibits a modest degree of regularity, a prior distribution is defined over a computationally expensive objective function by the means of a light-weight probabilistic emulator such as a GP. The constructed emulator is sequentially refined by adaptively sampling the next

training points based on acquisition functions derived from the posterior distribution. The trained emulator model is used to make predictions over the objective functions from the input space with minimum evaluation of the expensive true (simulator) function. Purely sampling-based iterative approaches (like genetic algorithms) are usually limited to drawing sparse random samples from proposals located nearby existing samples in the parameter space. In contrast, the use of predictive emulators permits exploration of the entire parameter space at higher resolution. This increases the chances of finding the true global optimum of the complex objective function in question and avoiding local optima.

Here, we use a single-layer Bayesian optimization approach to solve the multidimensional, multi-objective calibration of Open-Malaria (Fig. 1). Employing this single-layer Bayesian approach further allows for the direct comparison to previous calibration attempts for OpenMalaria as the objective functions are retained. We prove the strength and versatility of our approach by optimizing OpenMalaria's 23 input parameters using real-world data on 11 epidemiological outcomes in parallel. To emulate the solution space, we explore and compare two prior distributions, namely a GP emulator and a *superlearning* algorithm in form of a GP stacked generalization (GPSG) emulator. We first use a GP emulator to emulate the solution space. Whilst GP emulators provide flexibility whilst retaining relative simplicity[39] and have been used previously as priors in Bayesian optimization[38], stacked generalization algorithms have not. They provide a potentially attractive alternative as they have been shown to outperform GPs and other machine learning algorithms in capturing complex spaces[14,40]. The stacked generalization algorithm[40] builds on the idea of creating ensemble predictions from multiple learning algorithms (*level 0 learners*). The cross-validated predictions of the level 0 learners are incorporated into a general learning system (*level 1 meta-learner*). This allows for the combination of memory-efficient and probabilistic algorithms in order to reduce computational time, whilst retaining probabilistic elements required for adaptive sampling. Here, we showcase the efficiency and speed of the Bayesian optimization calibration scheme and propose a modus operandi to parameterize computationally intensive or complex mathematical models that harvests recent computational developments and is scalable to high dimensions in multi-objective calibration.

## Results

**Calibration workflow**. The developed model calibration workflow approach is summarized in Fig. 1a. In brief, goodness of fit scores were first derived for randomly generated, initial parameter sets. The goodness of fit scores were defined as a weighted sum of the loss functions for each of 11 fitting objectives. These span various epidemiological measures capturing the complexity and heterogeneity of the malaria transmission dynamics, including the age–prevalence and age–incidence relationships, and are informed by a multitude of observational studies (see the "Methods" section and Supplementary Note 2). Next, GP and GPSG emulators were trained on the obtained set of scores and used to approximate the relationship between parameter sets and goodness of fit for each objective. After initial investigation of different machine learning algorithms, the GPSG was constructed using a bilayer neural net, multivariate adaptive regression splines and random forest as *level 0* learners and a heteroscedastic GP as *level 1* learner (Fig. 1c, d, see the "Methods" section and supplement). Using a lower confidence bound acquisition function based on the emulators' point and uncertainty predictions for proposed new candidate parameter sets, the most promising sets were chosen. These parameter sets were simulated and added to the database of simulations for the next iteration of the algorithm.

At the next iteration, the emulators are re-trained on the new simulation database and re-evaluated (Fig. 1b). This iterative process of simulation, training and emulation was repeated until a memory limit of 1024 GB was hit. Approximately 130,000 simulations were completed up to this point.

**Algorithm performance (by iteration and time) and convergence**. Both emulators adequately captured the input–output relationship of the calculated loss-functions from the simulator, with better accuracy when close to minimal values of the weighted sum of the loss functions, $F$ (Fig. 2a). This is sufficient as the aim of both emulators within the Bayesian optimization framework is to find minimal loss function values rather than an overall optimal predictive performance for all outcome values. Examples of truth vs. predicted estimates on a 10% holdout set are provided in Fig. 2a (additional plots for all objectives can be found in Supplementary Figs. 2–5). A *satisfactory fit* of the simulator was previously defined by a loss function value of $F = 73.2$[21]. The *previous best* model fit derived using the GA had a weighted sum of the loss functions of $F = 63.7$[21]. *Satisfactory fit* was achieved by our approach in the first iteration of the GPSG-based Bayesian optimization algorithm (GPSG-BO), and after six iterations for the GP-based algorithm (GP-BO) (Fig. 2b). The *current best* fit was approximately retrieved after six iterations for the GPSG-BO algorithm and after nine iterations for GP-BO, and was improved by both algorithms after 10 iterations (returning final values $F = 58.3$ for GP-BO and 59.6 for GPSG-BO). This shows that the Bayesian optimization approach with either of our emulators very quickly achieves a better simulator fit than obtained with a classical GA approach that was previously employed to calibrate OpenMalaria. Of the two emulators, the GP approach finds a parameter set associated with a better overall accuracy and the GPSG reaches *satisfactory* values faster (both in terms of iterations and time). A likely explanation for this is that the GPSG-BO is unable to propagate its full predictive variance into the acquisition function. Only uncertainty stemming from the level 1 probabilistic learner (GP) is therefore captured in the final prediction. This leads to underestimation of the full predictive variance, and a bias towards exploitation in the early stages of the GPSG-BO algorithm (as illustrated by early narrow sampling, see Supplementary Figs. 6 and 7).

Figure 2c shows examples of the posterior estimates returned by the optimization algorithms in context of the log prior distributions for the parameters with the greatest effects on $F$ (see also Fig. 3c). All algorithms return parameter values within the same range and (apart from parameter 4), clearly distinct from the prior mean. The fact that highly similar parameter values are identified by multiple algorithms strengthens confidence in the final parameter sets yielded by the algorithms.

**Optimal goodness of fit**. The best fit parameter sets yielded by our approach are provided in Supplementary Table 2. Importantly, after ten iterations of the GPSG-BO algorithm (~7 days), and 20 iterations for the GP-BO algorithm (~12 days), both approaches yielded similar values of the 11 objective loss functions, along with similar weighted total loss function values, and qualitatively similar visual fits and predicted trends to the data (Fig. 3a, b and Supplementary material). We found this to be an unexpectedly fast result of the two algorithms. Details of the algorithm's best fits to the disease and epidemiological data are shown in Supplementary Figs. 8–18. Overall, several objectives had visual and reduced loss-function improvements, for example to the objective on the multiplicity of infection (Fig. 3a).

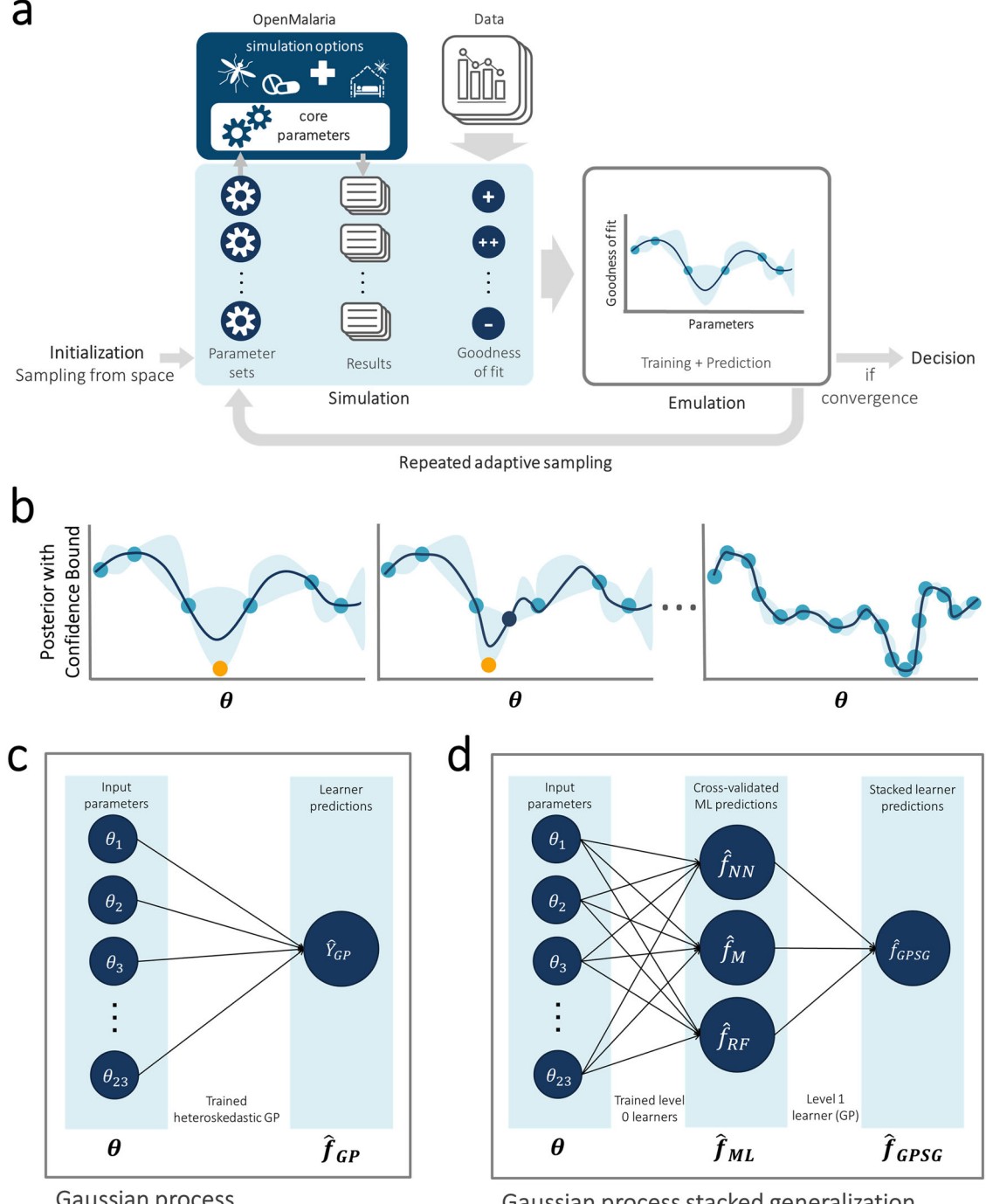

**Fig. 1 Overview of model calibration framework by Bayesian optimization, acquisition function, and Gaussian process and machine learning emulators.**
**a** General framework. The input parameter space is initially sampled in a space-filling manner, generating the initial core parameter sets (initialization). For each candidate set, simulations are performed with the model, mirroring the studies that yielded the calibration data. The deviation between simulation results and data is assessed, yielding goodness of fit scores for each parameter set. An emulator (**c** or **d**) is trained to capture the relationship between parameter sets and goodness of fit and used to generate out-of-sample predictions. Based on these, the most promising additional parameter sets are chosen (adaptive sampling by means of an acquisition function), evaluated, and added to the training set of simulations. Training and adaptive sampling are repeated until the emulator converges and a decision on the parameter set yielding the best fit is made. **b** Acquisition function. The acquisition function (black line) is used to determine new parameter space locations, $\theta$. $\theta$ is a vector of input parameters (23-dimensional for the model described here) to be evaluated during adaptive sampling (blue dot for previously evaluated locations, orange dot for new locations to be evaluated in the current iteration). It incorporates both predictive uncertainty (blue shading) of the emulator and proximity to the minimum. **c** Gaussian process (GP) emulator. A heteroscedastic Gaussian process is used to generate predictions on the loss functions, $\hat{f}_{GP}(\theta)$, for each input parameter set $\theta$. **d** Gaussian process stacked generalization (GPSG) emulator. Three machine learning algorithms (level 0 learners: bilayer neural net, multivariate adaptive regression splines and random forest) are used to generate predictions on the individual objective loss functions $\hat{f}_{NN}$, $\hat{f}_{M}$ and $\hat{f}_{RF}$ (collectively $\hat{f}_{ML}$) at locations $\theta$. These predictions are inputs to a heteroscedastic (level 1 learner) which is used to generate the stacked learner predictions $\hat{f}_{GPSG}$ and derive predictions on the overall goodness of fit $\hat{F}_{GPSG}$.

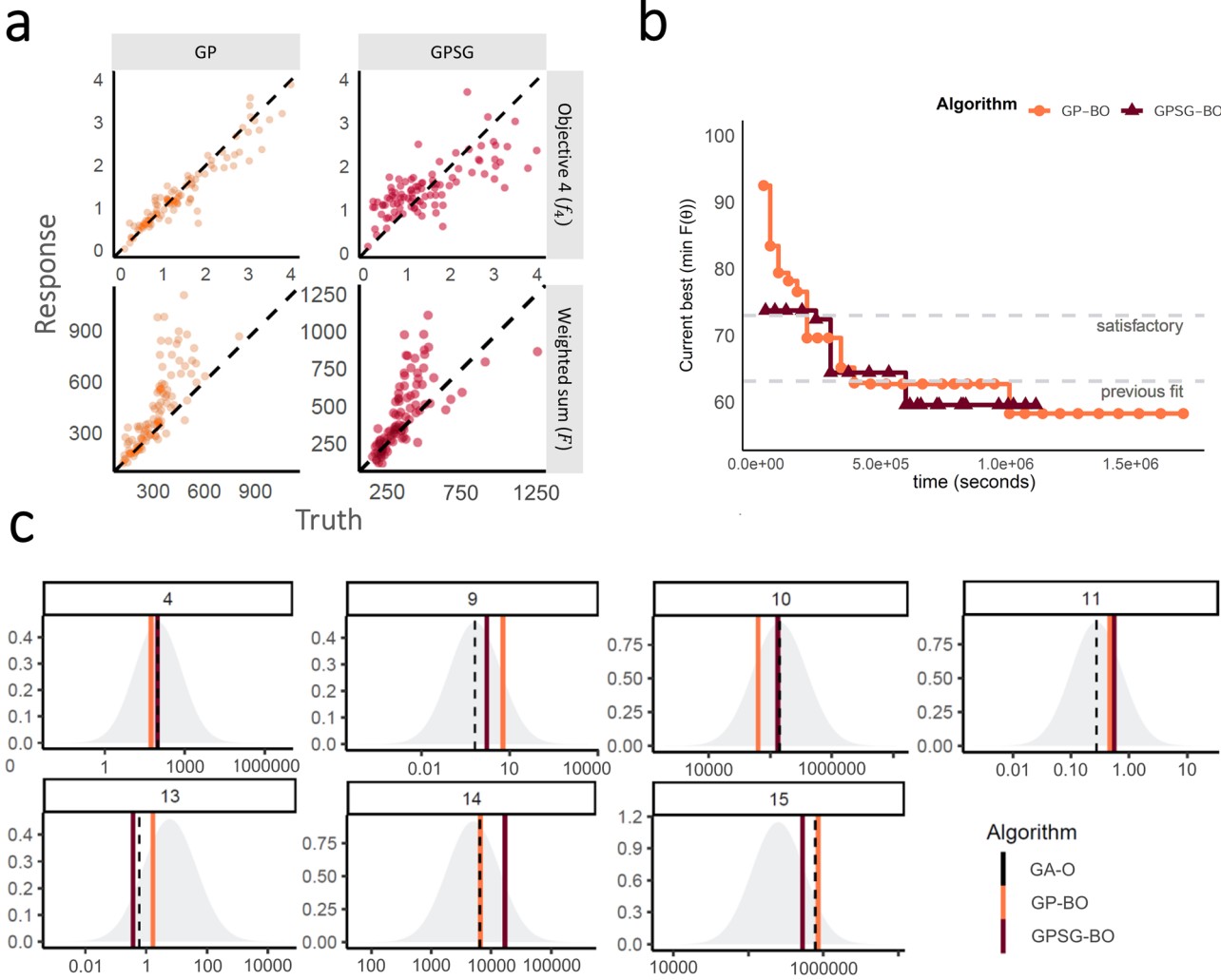

**Fig. 2 Emulator performance including predictions, convergence, and prior parameter distributions and posterior estimates. a** Example of emulator predictions vs. true values on a 10% holdout set. Predictions are shown for the final iteration of each optimization (orange dots for predictions in iteration 30 for GP-BO and red dots for predictions in iteration 23 for GPSG-BO). Here, emulator performances are shown for objective 4 (the age-dependent multiplicity of infection, $f_4$) and the weighted sum of loss functions over 11 objectives ($F(\theta)$). Plots for all other objectives are provided in the supplement. BO Bayesian optimization, GP Gaussian process emulator, GPSG Gaussian process stacked generalization emulator. **b** Convergence of the weighted sum of loss functions over 11 objectives ($F(\theta)$) associated with the current best fit parameter set by time in seconds. Satisfactory fit of OpenMalaria refers to a weighted sum of loss functions value of 73.2[21]. The previous best fit for OpenMalaria was achieved by the genetic algorithm and had a loss function value of 63.7. Our approach yields a fit $F$ of 58.2 for GP-BO in iteration 21 within $1.02e^6$ s (~12 days) and 59.6 for GPSG-BO in iteration 10 within $6.00e^5$ s (~7 days). GP-BO Gaussian process emulator Bayesian optimization, GPSG-BO Gaussian process stacked generalization emulator Bayesian optimization. **c** Example log prior parameter distributions (shown by the gray areas) and posterior estimates (vertical lines). The most influential parameters on the weighted sum of the loss functions are shown here in this figure (most influential parameters shown in Fig. 3c). All other plots can be found in the supplement. The posterior estimates for GP-BO (orange line) and GPSG-BO (red line) are shown in relation to those previously derived through optimization using a genetic algorithm (GA-O, dashed black line) for parameters $\theta_{4,9-11,13-15}$ (numbers in the panel labels).

**Impact/parameter sensitivity analysis and external validation.** An additional benefit of using emulators is the ability to understand the outcome's dependence on and sensitivity to the input parameters. To identify the most influential parameters for each of the 11 fitting objectives, we used the GP emulator trained on all available training simulation results from the optimization process ($R^2 = 0.53$ [objective 7]−0.92 [objective 3]) to conduct a global sensitivity analysis by variance decomposition (here via Sobol analysis[41]). Figure 3c shows Sobol total effect indices quantifying the importance of individual parameters and describing each parameter's contributions to the outcome variance for each objective. Our results indicate that most objectives are influenced by multiple parameters from different groups, albeit to varying degrees, thus highlighting the importance of

simultaneous multi-objective fitting. Clusters of influential parameters can be observed for most objectives; for example, parameters associated with incidence of acute disease influence clinical incidence and pyrogenic threshold objectives. Some parameters have strong influence on multiple objectives, such as parameter 4, the critical value of cumulative number of infections and influences immunity acquisition; and parameter 10, a factor required to determine the pyrogenic threshold, which we find to be a key parameter determining infections progressing to clinical illness.

*Algorithm validation.* To test if our algorithms can recover a known solution, the final parameter sets for both approaches were used to generate synthetic field data sets, and our approaches were

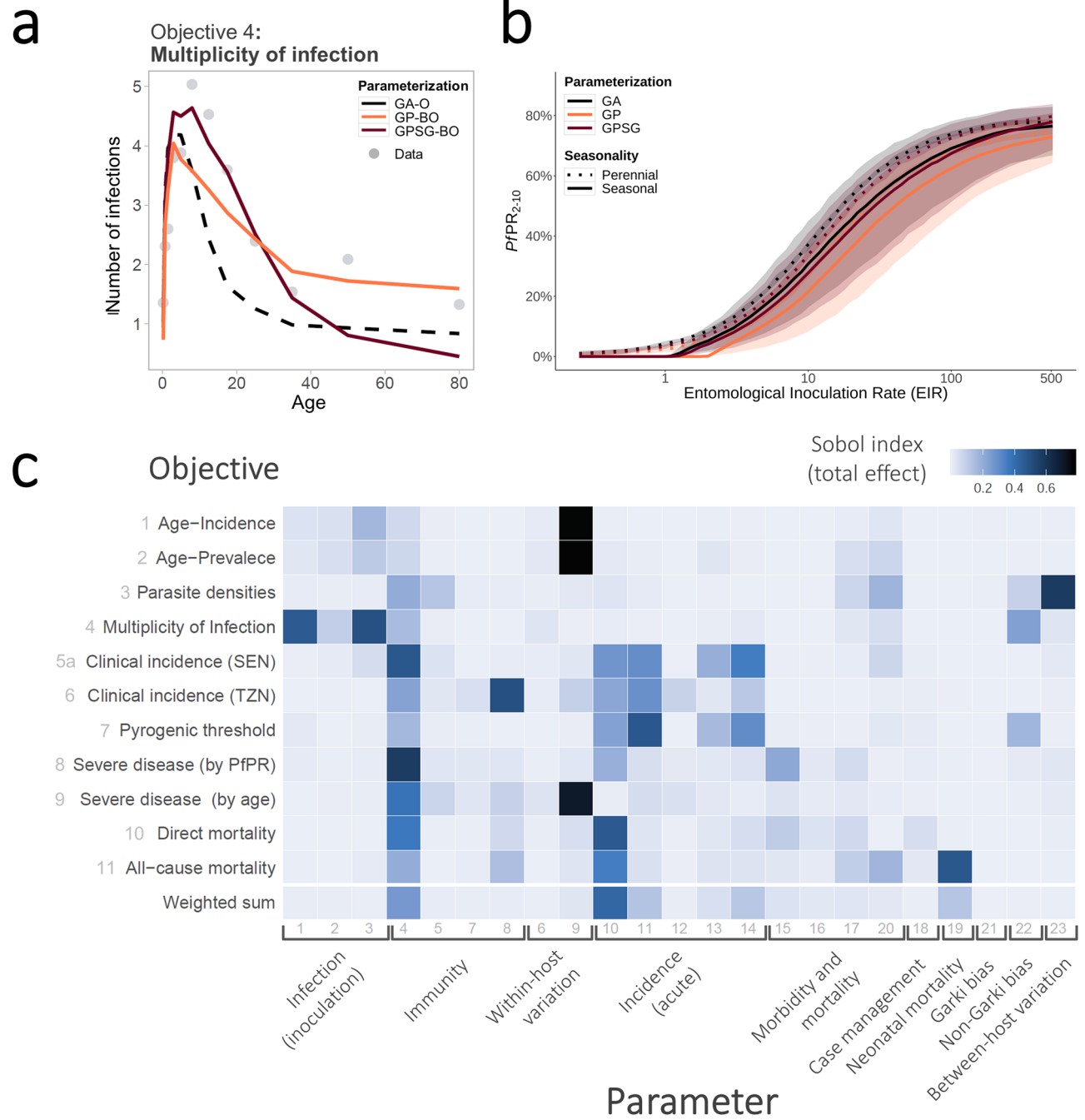

**Fig. 3 Exemplar plot of calibration and data for objective four "Multiplicity of infection", with exemplar epidemiological predictions of prevalence vs. EIR for the final calibration, and sensitivity of fitting objectives to each parameter. a** Multiplicity of infection by age. Comparison of simulator goodness of fit for objective 4, the age-specific multiplicity of infection (number of genetically distinct parasite strains concurrently present in one host). Simulations were carried out for the same random seed for all parameterizations and for a population size of $N = 5000$. **b** Simulated epidemiological relationship between transmission intensity (entomological inoculation rate, EIR) and *P. falciparum* prevalence rate ($PfPR_{2-10}$). Simulated epidemiological relationship between the transmission intensity (EIR in number of infectious bites per person per year) and infection prevalence in individuals aged 2–10 years ($PfPR_{2-10}$) under the parameterizations achieved by the different optimization algorithms. Lines show the mean across 100 random seed simulations for a simulated population size $N = 10,000$ and the shaded area shows the minimum to maximum range. **c** Parameter effects on the objective variance. Using the GP emulator, a global sensitivity analysis (Sobol analysis) was conducted. The tile shading shows the total effect indices for all objective functions and parameters grouped by function. SEN Senegal, TZN Tanzania.

subsequently applied to recover the known parameter set. For the GP, 13 of the 23 parameters were recovered (Supplementary Fig. 19a). Those not recovered largely represented parameters to which the weighted loss function was found to be insensitive (Fig. 3c). Thus, rather than showing a shortcoming of the

calibration algorithm, this suggests a potential for dimensionality reduction of the simulator and re-evaluation of its structure.

*Comparison of key epidemiological relationships and implications for predictions.* The new parameterizations for OpenMalaria were

further explored to assess key epidemiological relationships, in an approach similar multiple-model comparison in Penny et al. 2016[12]. We examined incidence and prevalence of disease, as well as incidence of mortality for multiple archetypical settings, considering a range of perennial and seasonal transmission intensity and patterns. The results are presented in Fig. 3b and Supplementary Figs. 20–30. The new parameterizations result in increased predicted incidence of severe episodes and decreased prevalence for all transmission intensities (thus also slightly modifying the prevalence–incidence relationship). While we found that the overall implications for the other simulated epidemiological relationship were small, the differences in predictions for severe disease may carry implications for public health decision-making and warrant further investigations. We conclude that our new parameterizations do not fundamentally bring into question previous research conducted using OpenMalaria, but we do suggest re-evaluation of adverse downstream events such as severe disease and mortality.

## Discussion

Calibrating IBMs can be challenging as many techniques struggle with high dimensionality, or become infeasible with long model simulation times and multiple calibration objectives. However, ensuring adequate model fit to key data is vital, as this impacts the weighting, we should give model predictions in the public health decision-making process. The Bayesian optimization approaches presented here provide fast solutions to calibrating IBMs while improving model accuracy, and by extension prediction accuracy.

Using a Bayesian optimization approach, we calibrated a detailed simulator of malaria transmission and epidemiology dynamics with 23 input parameters simultaneously to 11 epidemiological outcomes, including age-incidence and age-prevalence patterns. The use of a probabilistic emulator to predict goodness-of-fit, rather than conducting sparse sampling, allows for cheap evaluation of the simulator at many locations and increases our confidence that the final parameter set represents a global optimum. Our approach provides a fast calibration whilst also providing a better fit compared with the previous parameterization. We are further able to define formal endpoints to assess calibration alongside *visual confirmation* of goodness of fit[21,28], such as the emulator's predictive variance approaching the observed simulator variance. The emulator's ability to quantify the input stochasticity of the simulator also enables simulation at small population sizes, contributing to fast overall computation times.

Despite the demonstrated strong performance of stacked generalization in other contexts such as geospatial mapping[14,40,42–45], we found that using a *superlearning* emulator for Bayesian optimization was not superior to traditional GP-based methods. In our context using GPSG sped up convergence of the algorithm, but both approaches, GP and GPSG, led to equally good fits. Each approach does, however, have different properties with context-dependent benefits: The dimensionality reduction provided by GPSG approaches may lead to computational savings depending on the *level* 0 and *level* 1 learners. At the same time, only level 1 learner uncertainty is propagated into the final objective function predictions, which affects the efficacy of adaptive sampling and may lead to overly exploitative behavior, where sampling close to the point estimate of the predicted optimum is overemphasized, rather than exploring the entire parameter space (see Supplementary Tables S2 and S3 on selected points). On the other hand, exploration/exploitation trade-offs for traditional GP-BO algorithms have long been examined and *no regret* solutions have been developed[46].

The methodology presented here constitutes a highly flexible framework for individual based model calibration and aligns with the recent literature on using emulation in combination with stochastic computer simulation experiments of infectious diseases[47]. Both algorithms can be applied to other parameterization and optimization problems in disease modeling and also in other modeling fields, such as physical or mobility and transport models. Furthermore, in the GPSG approach, additional or alternative level 0 can be easily incorporated. Possible extensions to our approach include combination with methods to adaptively reduce the input space for constrained optimization problems[48], or other emulators may be chosen depending on the application. For example, homoscedastic GPs, which are faster than the heteroscedastic approach presented here, may be sufficient for many applications (but not for our IBM in which heteroscedastic was required due to the stochastic nature of the model). Alternatively, the computational power required by neural net algorithms scales only linearly (compared with a nominal cubic scaling for GPs) with the sample size, and we envisage wide applications for neural net-based Bayesian optimization in high dimensions. In our example, the bilayer neural net algorithm completed training and prediction within seconds whilst maintaining very high predictive performance. Unfortunately, estimating the uncertainty required for good acquisition functions is difficult in neural networks, but solutions are being developed[39,49]. These promising approaches should be explored as they become more widely available in high-level programming languages. With the increased availability of code libraries and algorithms, Bayesian optimization with a range of emulators is also becoming easier to implement.

The probabilistic, emulator-based calibration approach is accompanied by many benefits, including relatively quick global sensitivity analysis. As explored in this work, GP-based methods are easily coupled with sensitivity analyses, which provide detailed insights into a model's structural dependencies and the sensitivity of its goodness of fit to the input parameters. To the best of our knowledge, no other individual-based model calibration study has addressed this. In the case of malaria models, we have shown the interdependence of all OpenMalaria model components and a relative lack of modularity. In particular, within-host immunity-related parameters were shown to influence all fitting objectives, including downstream events such as severe disease and mortality when an infection progresses to clinical disease. Thus, calibrating within-host immunity in the absence of key epidemiology and population outcomes can lead to suboptimal calibration and ultimate failure of the model to adequately capture disease biology and epidemiology.

We have employed a different approach to calibrating OpenMalaria compared with previous methods but reach broadly similar comparisons to the natural history of disease. We also attained a slightly improved but similar goodness of fit, the main benefit being improved fitting times and the ability to measure parameter importance. Given the high number of influential parameters for each epidemiological objective in our parameter importance investigations, and the overlap between parameter–objective associations, we argue that, where possible, multi-objective fitting should be preferred over purely sequential approaches. Our approach confirms that using a parallel approach to parameterization rather than a modular, sequential, one captures the joint effects of all parameters and ensures that all outcomes are simultaneously accounted for. To the best of our knowledge, no model of malaria transmission of comparable complexity and a comparable number of fitting objectives was simultaneously calibrated to all its fitting objectives. Disregarding the joint influence of all parameters on the simulated outcomes may negatively impact the accuracy of model predictions, in particular on policy-relevant outcomes of severe disease and mortality.

Despite providing relatively fast calibration towards a better fitting parameter set, several limitations remain in our work. We have not systematically tested that a global optimum has been reached in our approach, but assume it is close to a global minimum for the current loss-functions defined, as further iterations did not yield changes, and both the GP and GPSG achieved similar weighted loss function and parameter sets. We aimed to improve the algorithm to calibrate detailed IBM, but we did not incorporate new data, which will be important moving forward as our parameter importance and validation analysis highlights several key epidemiological outcomes on severe disease and mortality are sensitive to results.

The key limitations of Bayesian optimization, particularly when using a GP emulator, are the high computational requirements in terms of memory and parallel computing nodes due to increasing runtimes and cubically scaling memory requirements of GPs. For this reason, we opted to not employ fully Bayesian KOH methods, which would double the number of GPs that would need to be run. Yet, memory limits may be reached before the predictive variance approached its limit. Furthermore, we chose an acquisition function with high probability to be *no regret*[46], but this likely overemphasizes exploration in the early stages of the algorithm considering the dimensionality of the problem and finite runtime. We opted here for pure exploitation every five iterations, but a more formal optimization of the acquisition function should be explored. The GPSG approach presented here can partially alleviate this challenge, depending on the choice of learning algorithms, but the iterative nature and need for many simulations remain. Memory-saving and time-saving extensions are thus worth exploring, such as incorporating graphics processing unit (GPU) computing or adaptively constraining the prior parameter space, dimensionality reduction, or addressing alternative acquisition functions. Additionally, as with all calibration methodologies, many choices are left to the user, such as the size of the initial set of simulations, the number of points added per iteration, or the number of replicates simulated at each location. There is no general solution to this as the optimal choices are highly dependent on the problem at hand, and we did not aim to optimize these. Performance might be optimized further through a formal analysis of all these variables, however the methodology here is already fast, effective, and highly generalizable to different types of simulation models and associated optimization problems. Improving the loss-functions or employing alternative *Pareto front* efficiency algorithms was not the focus of our current study but would be a natural extension of our work, as would be alternative approaches to the weighting of objectives, which remains a subjective component of multi-objective optimization problems[50].

A model's calibration to known input data forms the backbone of its predictions. The workflow presented here provides great advances in the calibration of detailed mathematical models of infectious diseases such as IBMs. Provided sufficient calibration data to determine goodness-of-fit, our approach is easily adaptable to any agent-based model and could become the modus operandi for multi-objective, high-dimensional calibration of stochastic disease simulators.

## Methods

**Preparation of calibration data and simulation experiments**. Disease transmission models generally have two types of parameter inputs: core parameters, inherent to the disease and determining how its natural history is captured, and simulation options characterizing the specific setting and the interventions in place (Fig. 1a). The simulation options specify the simulation context such as population demographics, transmission intensity, seasonality patterns, and interventions, and typically vary depending on the simulation experiment. In contrast, the core parameters determine how its epidemiology and aetiopathogenesis are captured. These include parameters for the description of immunity (e.g., decay of maternal

protection), or for defining clinical severe episodes (e.g., parasitemia threshold). To inform the estimation of core parameters, epidemiological data on the natural history of malaria were extracted from published literature and collated in previous calibrations of OpenMalaria[3,21,24], which were re-used in this calibration round and detailed in the Supplementary material. These include demographic data such as age-stratified numbers of host individuals which are used to derive a range of epidemiological outcomes such as age-specific prevalence and incidence patterns, mortality rates, and hospitalization rates.

Site-specific OpenMalaria simulations were prepared, representing the studies that yielded these epidemiological data in terms of transmission intensity, seasonal patterns, vector species, intervention history, case management, and diagnostics[24]. The mirroring of field study characteristics in the simulation options ensured that any deviation between simulation outputs and data could be attributed to the core parameters. Age-stratified simulation outputs to match to the data include numbers of host individuals, patent infections, and administered treatments. A summary of the data is provided in the Supplementary Note 2.

**General Bayesian optimization framework with emulators**. In our proposed Bayesian optimization framework (Fig. 1), we evaluated the deviation between simulation outputs and the epidemiological data by training probabilistic emulator functions that approximate the relationship between core parameter sets and goodness of fit. To test the optimization approach in this study we considered the original goodness of fit metrics for OpenMalaria detailed in ref. [21] and in Supplementary Note 2, which uses either residual sum of squares (RSS) or negative log-likelihood functions depending on the epidemiological data for each objective[21,24]. The objective function to be optimized is a weighted sum of the individual objectives' loss functions.

We adopted a Bayesian optimization framework where a probabilistic emulator function is constructed to make predictions over the loss functions for each objective from the input space, with a minimum amount of evaluations of the (computationally expensive) simulator.

We compared two emulation approaches. Firstly, a heteroskedastic GP emulator and secondly a stacked generalization emulator[40]. For approach 1 (GP-BO), we fitted a heteroskedastic GP with the input noise modeled as another GP[51] with a Matérn 5/2 kernel to account for the high variability in the parameter space (Fig. 1c)[38,52]. For approach 2 (GPSG-BO), we selected a two-layer neural network[53–55], multivariate adaptive regression splines[56], and a random forest algorithm[57,58] as level 0 learners.

With each iteration of the algorithm, the training was extended using adaptive sampling based on an acquisition function (lower confidence bound) that accounts for uncertainty and predicted proximity to the optimum of proposed locations (Fig. 1b). As the emulator performance improves (as assessed by its predictive performance on the test set) we gain confidence in the currently predicted optimum.

**Malaria transmission and disease simulator**. We applied our calibration approach to OpenMalaria (https://github.com/SwissTPH/openmalaria), an open-source modeling platform of malaria epidemiology and control. It features several related individual-based stochastic models of *P. falciparum* malaria transmission and control. Overall, the OpenMalaria IBM consists of a model of malaria in humans linked to a model of malaria in mosquitoes and accounts for individual level heterogeneity in humans (in exposure, immunity, and clinical progression) as well as aspects of vector ecology (e.g., seasonality and the mosquito feeding cycle). Stochasticity is featured by including between- and within-host stochastic variation in parasite densities with downstream effects on immunity[24]. OpenMalaria further includes aspects of the health system context (e.g., treatment seeking behavior and standard of care)[3,24] with additional probabilistic elements such as treatment seeking probabilities or the option for stochastic results of diagnostic tests. An ensemble of OpenMalaria model alternative variants is available defined by different assumptions about immunity decay, within-host dynamics, heterogeneity of transmission, along with more detailed sub-models that track parasite genetics, and pharmacokinetic and pharmacodynamics. The models allow for the simulation of interventions, such as the distribution of insecticide-treated nets (ITNs), vaccines, or reactive case detection[59,60], in comparatively realistic settings. Full details of the model and the history of calibration can be found in the original publications[3,21,24] and are summarized in Supplementary Notes 1 and 2. In our application, we use the term *simulator* to refer to the OpenMalaria base model variant[21].

**Calibrating OpenMalaria: loss functions and general approach**
*Aim*. Let $f(\theta)$ denote a vector of loss functions obtained by calculating the goodness of fit between simulation outputs and the real data (full details of loss function can be found in supplementary Note 2). In order to ensure a good fit of the model, we aim to find the parameter set $\theta$ that achieves the minimum of the weighted sum of 11 loss functions (corresponding to the 10 fitting objectives) $F(\theta) = \sum_{i=1}^{11} w_i f_i(\theta)$, where $f_i(\theta)$ is the value of objective function $i$ at $\theta$ and $w_i$ is the weight assigned to

objective function $i$:

$$\underset{\theta}{\arg\min}\left(\sum_{i=1}^{11} w_i f_i(\boldsymbol{\theta})\right) \tag{1}$$

The weights are kept consistent with previous rounds of calibration and chosen such that different epidemiological quantities contributed approximately equally to $F(\boldsymbol{\theta})$ (see Supplementary Note 2).

## Step 1: Initialization

Let $D = 23$ denote the number of dimensions of the input parameter space $\boldsymbol{\Theta}$ and $W = 11$ the number of objective functions $f_i(\boldsymbol{\theta}), i = 1, \ldots, 11$. Prior distributions consistent with previous fitting runs[21] were placed on the input parameters. As each parameter is measured in different units, we sampled from the $D$-dimensional unit cube $\boldsymbol{\Theta}$ and converted these to quantiles of the prior distributions[21] (Supplementary Note 2 and Supplementary Fig. 6). Previous research suggests that in high-dimensional spaces quasi-Monte Carlo (qMC) sampling outperforms random or Latin Hypercube designs for most function types and leads to faster rates of convergence[61,62]. We therefore used Sobol sequences to sample 1000 initial locations from $\boldsymbol{\Theta}$. The GP can account for input stochasticity of the simulator. For each sample, we simulated 2 random seeds at a population size of 10,000 individuals. Additionally, 100 simulations were run at the centroid location of the unit cube to gain information on the simulator noise. Using small noisy simulations with small populations speeds up the fitting as the noisy simulations are less computational expensive than larger population runs. Replicates were used to detect signals in noisy settings and estimate the pure simulation variance[51]. The 2000 unique locations were randomly split into a training set (90%) and a test set (10%). All simulator realizations at the centroid were added to the training set.

## Step 2: Emulation

**Emulator training** Each emulator type for each objective function was trained in parallel to learn the relationships between the normalized input space $\boldsymbol{\Theta}$, and the log-transform of the objective functions $f(\boldsymbol{\theta})$. In each dimension $dD$, the mean $\mu_d$ and standard deviation $\sigma_d$ of the training set were recorded, $d = 1, \ldots, 23$.

**Posterior prediction** We randomly sampled 500,000 test locations in $\boldsymbol{\Theta}$ from a multivariate normal distribution with mean $\boldsymbol{\theta}_{opt}$ and covariance matrix $\boldsymbol{\Sigma}$, where $\boldsymbol{\theta}_{opt}$ is the location of the current best location and $\boldsymbol{\Sigma}$ is determined based on previously sampled locations, and scaled each dimension to mean $\mu_d$ and standard deviation $\sigma_d$. The trained emulators were used to make predictions $F(\boldsymbol{\theta})$ of the objective functions $F(\boldsymbol{\theta})$ at the test locations. Mean estimates, standard deviations, and nugget terms were recorded. The full predictive variance at each location $\boldsymbol{\theta}\Theta$ corresponds to the sum of the standard deviation and nugget terms. From this, we derived the weighted sum

$$\hat{F}(\boldsymbol{\theta}) = \sum_{i=1}^{11} w_i f_i(\boldsymbol{\theta}), \tag{2}$$

using weights $\boldsymbol{w}$ consistent with previous fitting runs[63] with greater weighting for further downstream objectives. The predicted weighted loss function at location $\boldsymbol{\theta}$ was denoted $\hat{F}(\boldsymbol{\theta})$ with a predicted mean $\hat{\mu}_F(\boldsymbol{\theta})$ and variance $\hat{\sigma}_F(\boldsymbol{\theta})$. Every 15 iterations, we increase the test location sample size to 5 million to achieve denser predictions.

## Step 3: Acquisition

We chose the lower confidence bound (LCB) acquisition function to guide the search of the global minimum[64]. Lower acquisition corresponds to *potentially* low values of the weighted objective function, either because of a low mean prediction value or large uncertainty[65]. From the prediction set at iteration $t$, we sample without replacement 250 new locations

$$\boldsymbol{\theta} = \arg\min_{\boldsymbol{\theta}}\{\hat{\mu}_F(\theta, t) - \sqrt{\nu \tau_t}\hat{\sigma}_t(\theta, t)\} \tag{3}$$

with the hyperparameter $\nu = 1$ and

$$\tau_t = 2\log(T_t^{D/2+2}\pi/3\partial), \tag{4}$$

where $T_t$ is the number of previous unique realizations of the simulator at iteration $t$, and $\delta = 0.01$ is a hyperparameter[46]. We choose this method as with high probability it is *no regret*[46,65]. With increasing iterations, confidence bound-based methods naturally transition from mainly exploration to exploitation of the current estimated minimum. In addition to this, we force exploitation every 10 iterations by setting $T_t = 0$.

## Step 4: Simulate

The simulator was evaluated at locations identified in step 3 and the realizations were added to the training set. Steps 2–4 were run iteratively. The Euclidian distance between locations of current best realizations was recorded.

## Step 5: Convergence

Convergence was defined as no improvement in the best realization, $\arg\min_F F$.

**Emulator definition**. We compared two emulation approaches. Firstly, a heteroskedastic GP emulator and secondly a stacked generalization emulator[40] using a two-layer neural net, multivariate adaptive regression splines (MARS) and a random forest as level 0 learners and a heteroskedastic GP as level 1 learner:

*Heteroskedastic Gaussian Process (hetGP)*[66]. We fitted a GP with the input noise modeled as another GP[51]. After initial exploration of different kernels, we chose a Matérn 5/2 kernel to account for the high variability in the parameter space. A Matérn 3/2 correlation function was also tested performed equally. Each time the model was built (for each objective at each iteration), its likelihood was compared to that of a homoscedastic GP and the latter was chosen if its likelihood was higher. This resulted in a highly flexible approach, choosing the best option for the current task.

*GP stacked generalization*. Stacked generalization was first proposed by Wolpert 1992[40] and builds on the idea of creating ensemble predictions from multiple learning algorithms (level 0 learners). In *superlearning*, the cross-validated predictions of the level 0 learners are fed into a level 1 meta-learner. We compared the 10-fold cross-validated predictive performance of twelve machine learning algorithms on the test set. All algorithms were accessed through the mlr package in R version 2.17.0[67]. We compared two neural network algorithms (brnn[54] for a two-layer neural network and nnet for a single-hidden-layer neural network[68]), five regression algorithms (cvglmnet[69] for a generalized linear model with LASSO or Elasticnet Regularization and 10-fold cross validated lambda, glmboost[70] for a boosted generalized linear model, glmnet[69] for a regular GLM with Lasso or Elasticnet regularization, mars for multivariate adaptive regression splines[71], and cubist for rule-and instance-based regression modeling[72]), three random forest algorithms (randomForest[58], randomForestSRC[73], and ranger[74]), and a tree-like node harvesting algorithm (nodeHarvest[75]). Extreme gradient boosting and support vector regression were also tested but excluded from the comparison due to its long runtime. Their performance was compared with regards to runtimes, and correlation coefficients between predictions on the test set and the true values. Based on these, we selected the two-layer neural network (brnn[55]), multivariate adaptive regression spline (mars[71]), and random forest (randomForest[58]) algorithms. This ensemble of machine learning models constituted the level 0 learners and was fitted to the initialization set. Out-of-sample predictions from a 10-fold cross validation of each observation were used to fit the level 1 heteroskedastic GP. As in approach 1, we opted for a Matérn 5/2 kernel and retained the option of changing to a homoscedastic model where necessary.

**Emulator performance**. We ascertained that both emulators captured the input-output relationship of the simulator by tracking the correlation between true values $f$ and predicted values $\hat{f}$ on the holdout set of 10% of initial simulations with each iteration (truth vs. predicted $R^2 = 0.51$–$0.89$ for GP vs. $0.37$–$0.77$ for GPSG after initialization, see Supplementary Table 1). Transition from exploration to exploitation during adaptive sampling was tracked by recording the distribution of points selected during adaptive sampling in each iteration (Supplementary Figs. 2 and 3).

**Sensitivity analysis**. A global sensitivity analysis was conducted on a heteroskedastic GP model with Matérn 5/2 kernel that was trained on all training simulation outputs ($n = 5400$) from the fitting process. We used the Jansen method of Monte Carlo estimation of Sobol' sensitivity indices for variance decomposition[76,77] with 20,000 sample points and 1000 bootstrap replicates. Sobol' indices were calculated for all loss functions $f$ as well as for their weighted sum $F$ and in all dimensions. Whilst keeping the number of sample points as low as possible for computational reasons, we ascertained that first-order indices summed to 1 and total effects >1. We further ensured that the overall results of the Sobol' analysis were consistent with the results of other global sensitivity analyses, namely the relative parameter importance derived from training a random forest (Supplementary Fig. 32).

**Synthetic data validation**. Synthetic field data was generated by forward simulation using the final parameter sets from each optimization process. The two optimization algorithms were run anew using the respectively generated synthetic data to calculate the goodness of fit statistics. The parameter sets retrieved by the validation were compared against the parameterization yielded by the optimization process.

**Epidemiological outcome comparison**. We conducted a small experiment to compare key epidemiological outcomes from the new parameterizations with the original model and that detail in a four malaria model comparison in Penny et al. 2016[12]. We simulated malaria in archetypical transmission and seasonality settings using the different parameterizations. The experiments were set up in a full-factorial fashion, considering the simulation options described in Table 1. Monitored outcomes were the incidence of uncomplicated, severe disease, hospitalizations, and indirect and direct malaria mortality over time and by age, prevalence over time and by age, the prevalence–incidence relationship, and the EIR–prevalence relationship. Simulations were conducted for a population of 10,000 individuals over 10 years.

**Table 1 Full experimental design in setting archetypes.**

| Number of stochastic realizations | Seasonality | Transmission (EIR) | Parameterization |
|---|---|---|---|
| 10 | Perennial Seasonal (sinusoidal) | 0.25, 0.5, 0.75, 1, 1.1, 1.25, 1.35, 1.5, 1.75, 2, 2.5, 3, 4, 5, 6, 7, 8, 9, 10, 12, 14, 16, 18, 20, 22, 25, 30, 35, 40, 45, 50, 64, 73, 80, 100, 128, 150, 200, 256, 512 | GA GP-BO GPSG-BO |

Experiments were run at 36% probability that an infected individual with clinical symptoms receives effective treatment within 14 days.

**Software**. Consistent with previous calibration work, we used OpenMalaria version 35, an open-source simulator written in C++ and further detailed in full in the supplement, as well as OpenMalaria wiki (https://github.com/SwissTPH/openmalaria/wiki) or in the original publications[3,21,24]. Calibration was performed using R 3.6.0. For the machine learning processes, all algorithms were accessed through the mlr package version 2.17.0[67]. The heteroskedastic GP utilized the hetGP package under version 1.1.2[66]. The sensitivity analysis was conducted using the soboljansen function of the sensitivity package version 1.21.0 in R[78]. All algorithms were adapted to the operating system (CentOS 7.5.1804) and computational resources available at the University of Basel Center for Scientific Computing, SciCORE, which uses a Slurm queueing system. The full algorithm code is available on GitHub and deposited in the zendo database under accession code https://doi.org/10.5281/zenodo.5595100 and can be easily adapted to calibrate any simulation model. The number of input parameters and objective functions are flexible. Thus, to adapt the code to other simulators, code should be updated to run the respective model simulator, and tailored to user's operating system. Further requirements to adapt the workflow are sufficient calibration data, and a per-objective goodness-of-fit metric.

**Reporting summary**. Further information on research design is available in the Nature Research Reporting Summary linked to this article.

## Data availability
All calibration data are detailed in ref. [24]. The data used for model fitting are available on GitHub and deposited in the zendo database under accession code https://doi.org/10.5281/zenodo.5595100. The data generated in this study and plotted in the main manuscript or supplement are publicly available and have been deposited in the zendo database under accession code https://doi.org/10.5281/zenodo.5552279.

## Code availability
Code is publicly available on GitHub and deposited in the zendo database under accession code https://doi.org/10.5281/zenodo.5595100.

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

## Acknowledgements

We acknowledge and thank our colleagues in the Swiss TPH Disease Modeling unit. Calculations were performed at sciCORE (http://scicore.unibas.ch/) scientific computing core facility at University of Basel. The work was funded by the Swiss National Science Foundation through SNSF Professorship of M.A.P. (PP00P3_170702) supporting M.A.P., M.G., and L.B. T.R. was supported by Bill & Melinda Gates Foundation Project OPP1032350 to T.A.S. EC's research is supported by funding from the Bill and Melinda Gates Foundation to Curtin University (Opportunity ID: OPP1197730).

## Author contributions

M.A.P. and E.C. conceived the study. Algorithm development by E.C., T.R., M.A.P., and S.F. with implementation and preparation for sharing on GitHub by T.R. Loss functions by M.A.P. and T.A.S. Sensitivity analysis by T.R. with inputs from M.G. and L.B. First draft was written by T.R. and M.A.P., all authors contributed to writing and interpretation of results and approved the final manuscript.

## Competing interests

The authors declare no competing interests.
