## [Peer Review File · Nature Communications]

Reviewers' Comments:

Reviewer #1:

Remarks to the Author:

NCOMMS-21-14090-T

Machine learning approaches to calibrate individual-based infectious disease models

SUMMARY

I like this paper. The authors might not realize it, but their contribution is on the frontier of what Baker et al (<https://arxiv.org/abs/2002.01321>) describe for stochastic computer simulation experiments. Baker explicitly mentions high-dimensional inputs and multiple outputs as very hard. We need more like this, as simulation-based computing continues to democratize scientific modeling and data analytics. That said, I have some thoughts that might help connect methods in this paper to that literature.

I see this paper in the context of stochastic simulation generally, of which IBMs are a very important example. (Baker, et al. discuss a similar Ebola simulation and calibration example.) There are two important modern textbooks that cover this area, one by Santner Williams and Notz (SWN, <https://www.springer.com/gp/book/9781493988457>) and another by Gramacy (G, <https://bobby.gramacy.com/surrogates/>). Using chapters/sections from those books, the contribution in the current manuscript may be described as follows.

A surrogate model or emulator (SWN Ch4, or G Ch5) for stochastic simulations (G Ch10) is used to frame calibration (SWN Ch8, or G Ch8.1) for multiple objectives (novel) as a Bayesian optimization (SWN Ch6, or G Ch7) problem and for sensitivity analysis (SWN Ch7, or G Ch8.2). Framing calibration as optimization isn't novel, but I haven't seen it for multiple objectives. The authors also propose a cascade of "machine learning" models for their emulator which is also, in a way, novel. I'll have something to say about both of these below. Their application is quite high dimensional compared to most, which makes things very challenging. The authors approach seems to work well!

MAJOR POINT

Even though I like the methodological and applied contributions in this paper, I am concerned that the authors are not aware of the state-of-the-art in surrogate modeling and calibration of (stochastic) simulations. Consequently, they fail to connect with that literature and miss opportunities to contrast with those approaches, and to an extent they also reinvent the wheel to some

extent.

The canonical (Bayesian) calibration apparatus from those textbooks (SWN Ch7, or G Ch8.1) is due to Kennedy & O'Hagan (2001, KOH). It does not involve optimizing a discrepancy, and this is deliberate. It involves a multi-level cascade of surrogates (usually GPs, but it doesn't need to be), like your GPSSG, to account for bias in the computer simulations relative to the field experiment. So this begs the question: why didn't you take this approach, and what is different/better about what you did?

I think you can make a case for KOH being too cumbersome in your setting, with many competing objectives and with a high-dimensional parameter space. However, I still think that the problem involves many elements that are "textbook", as summarized above, but no connection is being made. For example, Exercise 10.4.5 in the G book asks students to calibrate a stochastic simulator in 4 input dimensions. Some of my students were recently able to do get results similar to the solution key provided by the author. Could these be scaled up to more parameters and more objectives? Possibly; although I'm not saying it would be easy!

MINOR POINTS

I don't like the "machine learning approaches ..." in the title. This paper isn't really about machine learning methods. You might use some neural networks in your GPSSG, but it could be anything really. And you end up criticizing that choice (see below). The canonical KOH setup uses multiple GPs. Also, "infectious disease models" is too generic. I would focus on malaria and the unique challenges of that application, including high input dimension and multiple objectives.

I don't like the comparison to genetic algorithms. It may be that folks are using them for calibration in some settings. But nobody who has worked on calibration of computer experiments (e.g., those books) would do that, at least not in the last 20 years. It's too weak of a straw man for all the reasons you say.

On p11 near line 255 you say "However, this was beyond the scope of the current work to develop fast and powerful calibration methodology for IBMs."

What? I thought that's what this paper was about!

On p13 near line 315 you say "Unfortunately, estimating the uncertainty required for good acquisition functions is difficult in neural networks ...".
yes! This is why the computer simulation experiments community prefers

GPs.

This is why KOH built their multi-level cascade from GPs. This is why you shouldn't emphasize "machine learning approaches" in the title.

On p18 near line 455 you say "Computational savings were later achieved through pre-averaging of replicates". If you're describing what hetGP does, then this is not quite correct. It does work with a reduced set of sufficient statistics, some of which are averages over replicates, but it is not just a matter of pre-processing.

On p19 near line 480. For some reason you have switched to a different citation style here. Also, I think you are missing a cite for LCB earlier in the paragraph. Or is this that cite? You could also benefit from a general BO cite.

Similarly strange citations on p22 near line 555.

Please cite R packages. See, e.g., `citation("hetGP")` or `citation("hetGP")` for suggestions from the authors

Reviewer #2:

Remarks to the Author:

This manuscript proposes an advance on the methods by which agent-based models of infectious diseases are fitted to data by using machine learning emulators of the model's goodness of fit to data. This approach offers a very sensible shortcut around the computational bottleneck of performing many replicate simulations in an iterative fitting process. Given the novelty of this approach and the clear and excellent demonstration of it on a well-established and influential agent-based model of malaria, this manuscript represents an important contribution to the literature on agent-based modeling of infectious diseases.

(line 42) It's strange that references are provided for all the diseases mentioned individually except dengue. One option could be <https://journals.plos.org/ploscompbiol/article?id=10.1371/journal.pcbi.1006710>

(line 64) This context about the number of stars in the observable Universe is fantastic.

(lines 400-401) The choice of weights used for the multiple loss functions is not clear. It would be good to have a better idea of how they were chosen. This is a serious issue given the authors' recommendation in the discussion (around line 334) that multi-objective fitting should be preferred over purely sequential approaches. That's much easier to do when the weights for the multiple objectives are pre-specified and taken on faith as appropriate.

(lines 403-404) It would be helpful if more could be said about the minimum number of evaluations needed to inform the emulator function. Convergence is mentioned on line 413, but this is rather vague.

(line 458) It might be useful to add a sentence with a bottom-line statement of the total number of model simulations required for this calibration procedure.

(line 188 and on) The results about performance are quite interesting. I would expect faster performance than previous methods, but a better fit overall is a nice cherry on top. Well done.

(line 194) What is a "satisfactory" fit? Something a little bit worse than the fit obtained? I'll have to remember this trick! ;)

(line 238) Missing word to begin the paragraph?

(discussion) Any more commentary that the authors might be able to provide for how their approach could be applied to other agent-based models would be valuable. Are there unique aspects of OpenMalaria and its well trodden principles for fitting to data that make this approach more successful here than it might be for other models?

Alex Perkins

Response to Reviewers for

Title: (original) *Machine learning approaches to calibrate individual-based infectious disease model*

Authors: Theresa Reiker^{1,2}, Monica Golumbeanu^{1,2}, Andrew Shattock^{1,2}, Lydia Burgert^{1,2}, Thomas A. Smith^{1,2}, Sarah Fillipi³, Ewan Cameron^{4,5,6}, Melissa A. Penny^{1,2*}.

We would like to thank the Editor and the two Reviewers for considering our manuscript. We highly appreciate their critical assessment of our work and useful and constructive comments which has strengthened our paper. We would like to thank the reviewers for the detailed pointers to specific additional references and the broader literature. We have thus followed further details in the literature and made modifications to the manuscript in response.

In the following text we address the editor and reviewer remarks point by point (reviewers remarks are in black text and our responses in blue). Modifications to the manuscript as marked by approximate line number and included in "*italic*" here if needed.

Response to reviewers' feedback

Reviewer 1

SUMMARY

I like this paper. The authors might not realize it, but their contribution is on the frontier of what Baker et al (<https://arxiv.org/abs/2002.01321>) describe for stochastic computer simulation experiments. Baker explicitly mentions high-dimensional inputs and multiple outputs as very hard. We need more like this, as simulation-based computing continues to democratize scientific modeling and data analytics. That said, I have some thoughts that might help connect methods in this paper to that literature.

I see this paper in the context of stochastic simulation generally, of which IBMs are a very important example. (Baker, et al. discuss a similar Ebola simulation and calibration example.) There are two important modern textbooks that cover this area, one by Santner Williams and Notz (SWN, <https://www.springer.com/gp/book/9781493988457>) and another by Gramacy (G, <https://bobby.gramacy.com/surrogates/>). Using chapters/sections from those books, the contribution in the current manuscript may be described as follows.

A surrogate model or emulator (SWN Ch4, or G Ch5) for stochastic simulations (G Ch10) is used to frame calibration (SWN Ch8, or G Ch8.1) for multiple objectives (novel) as a Bayesian optimization (SWN Ch6, or G Ch7) problem and for sensitivity analysis (SWN Ch7, or G Ch8.2). Framing calibration as optimization isn't novel, but I haven't seen it for multiple objectives. The authors also propose a cascade of "machine learning" models for their emulator which is also, in a way, novel. I'll have something to say about both of these below. Their application is quite high dimensional compared to most, which makes things very challenging. The authors approach seems to work well!

We thank Reviewer 1 for their positive assessment and comments below. We appreciate placing our work in context Baker et al. and pointing out the challenge with multiple

objectives which was indeed both the motivator for our approach and key challenge we hope our approach can address for other models. Your suggestions have supported improvement of our paper, thank you.

In response we have added a reference to this manuscript in the discussion (line 375)

The methodology presented here constitutes a highly flexible framework for individual based model calibration and aligns with the recent literature on using emulation in combination with stochastic computer simulation experiments of infectious diseases (47).

Other point by point responses are below

MAJOR POINT

Even though I like the methodological and applied contributions in this paper, I am concerned that the authors are not aware of the state-of-the-art in surrogate modeling and calibration of (stochastic) simulations. Consequently, they fail to connect with that literature and miss opportunities to contrast with those approaches, and to an extent they also reinvent the wheel to some extent.

The canonical (Bayesian) calibration apparatus from those textbooks (SWN Ch7, or G Ch8.1) is due to Kennedy & O'Hagan (2001, KOH). It does not involve optimizing a discrepancy, and this is deliberate. It involves a multilevel cascade of surrogates (usually GPs, but it doesn't need to be), like your GPSG, to account for bias in the computer simulations relative to the field experiment. So this begs the question: **why didn't you take this approach, and what is different/better about what you did?**

I think you can make a case for KOH being too cumbersome in your setting, with many competing objectives and with a high-dimensional parameter space. However, I still think that the problem involves many elements that are "textbook", as summarized above, but no connection is being made. For example, Exercise 10.4.5 in the G book asks students to calibrate a stochastic simulator in 4 input dimensions. Some of my students were recently able to do get results similar to the solution key provided by the author. Could these be scaled up to more parameters and more objectives? Possibly; although I'm not saying it would be easy!

We thank the reviewer for this comment, specifically for highlighting we need to place the work more solidly in the context of current literature. Thank you for these specific pointers to key papers. We consolidated the references mentioned by the reviewer, and along with some additional references we modified text and adapted literature citations in our manuscript. We specify the following reasons for not using KOH:

1. Computational resources. The KOH approach would require double the number of Gaussian processes to be fit. As it stands, the GP memory requirements were the limiting factor in the current execution of the calibration. We believe that under the current implementation framework (that is programmed in R and using a computing cluster that does not allow for server connection and thereby doesn't allow connecting to cloud based AutoML platforms such as h20), scaling to a 23-dimensional space with 11 objective functions would not be feasible.
2. Comparability and interpretability. In retaining the existing overall calibration framework of OpenMalaria with its explicit simulations and loss function calculation, we were able draw direct comparisons and examine the influence of each parameter

on our 11 objectives. Analysing the parameter influence on each objective loss-function also supports understanding model and epidemiological relationships. These insights allowed us to make clear interpretations of the effects of data scarcity and its biological and epidemiological implications. These will be used in future model development

We have added a section to the manuscript (around lines 135) linking to KOH and explaining our reasoning for using a single-layer Bayesian optimization approach.

Progress in recent years on numerical methods for supervised, regularized learning of smooth functions from discrete training data allows us to revisit calibration of detailed mathematical models using Bayesian methods for global optimization (34). Current state-of-the art calibration approaches for stochastic simulators are often based around Kennedy and O'Hagan's approach (35) (KOH), where a posterior distribution for the calibration parameters is derived through a two-layer Bayesian approach involving cascade of surrogates (usually GPs) (36). A first GP is used to model the systematic deviation between the simulator and the real process it represents, while a second GP is used to emulate the simulator (37). However, this approach is computationally intense when scaling to high-dimensional input spaces and multi-objective optimization. A fully Bayesian KOH approach is likely computationally heavy (37) for the efficient calibration of detailed malaria simulators like OpenMalaria. Single-layer Bayesian optimization with Gaussian processes (GPs) on the other hand have gained popularity as an efficient approach to tackle expensive optimization problems, for example in hyperparameter search problems in machine learning (38, 39).

MINOR POINTS

I don't like the "machine learning approaches ..." in the title. This paper isn't really about machine learning methods. You might use some neural networks in your GPSG, but it could be anything really. And you end up criticizing that choice (see below). The canonical KOH setup uses multiple GPs.

Thank you for this comment. In context we have decided to adapt the title to

Emulator-based Bayesian optimization for efficient multi-objective calibration of an individual-based model of malaria

Also, "infections disease models" is too generic. I would focus on malaria and the unique challenges of that application, including high input dimension and multiple objectives.

Thank you for this point, we added text along this focus around line 80

Malaria IBMs in particular are often highly complex (e.g. containing multiple sub-modules and many parameters), consider a two-host system influenced by seasonal dynamics, and often account for multifaceted within-host dynamics.

I don't like the comparison to genetic algorithms. It may be that folks are using them for calibration in some settings. But nobody who has worked on calibration of computer

experiments (e.g., those books) would do that, at least not in the last 20 years. It's too weak of a straw man for all the reasons you say.

We shortened this comparison (lines 90), and reduced throughout.

On p11 near line 255 you say "However, this was beyond the scope of the current work to develop fast and powerful calibration methodology for IBMs." What? I thought that's what this paper was about!

Apologies for this misleading wording it was a rudiment from a previous version of the manuscript. We have now deleted the sentence.

On p13 near line 315 you say "Unfortunately, estimating the uncertainty required for good acquisition functions is difficult in neural networks ...". yes! This is why the computer simulation experiments community prefers GPs. This is why KOH built their multi-level cascade from GPs. This is why you shouldn't emphasize "machine learning approaches" in the title.

We adapted our title as above

On p18 near line 455 you say "Computational savings were later achieved through pre-averaging of replicates". If you're describing what hetGP does, then this is not quite correct. It does work with a reduced set of sufficient statistics, some of which are averages over replicates, but it is not just a matter of pre-processing.

Thank you for this catch, we have now deleted this sentence

On p19 near line 480. For some reason you have switched to a different citation style here.

Apologies for this, we have now updated and corrected all references

Also, I think you are missing a cite for LCB earlier in the paragraph. Or is this that cite?

We have now corrected Srinivas 2009, and added Auer 2002 (P. Auer, Using confidence bounds for exploitation-exploration trade-offs. *Journal of Machine Learning Research* 3, 397-422 (2002).)

You could also benefit from a general BO cite.

Thank you: we now cite Mockus 1989 added in line 98

Similarly strange citations on p22 near like 555.

We updated all references, thank you.

Please cite R packages. See, e.g., citation("hetGP") or citation("hetGP") for suggestions from the authors

Thank you we have now added citations

Reviewer 2

This manuscript proposes an advance on the methods by which agent-based models of infectious diseases are fitted to data by using machine learning emulators of the model's goodness of fit to data. This approach offers a very sensible shortcut around the computational bottleneck of performing many replicate simulations in an iterative fitting process. Given the novelty of this approach and the clear and excellent demonstration of it on a well-established and influential agent-based model of malaria, this manuscript represents an important contribution to the literature on agent-based modeling of infectious diseases.

We thank Reviewer 2 for their review and positive comments as well as suggested changes below. Thank you!

(line 42) It's strange that references are provided for all the diseases mentioned individually except dengue. One option could be <https://journals.plos.org/ploscompbiol/article?id=10.1371/journal.pcbi.1006710>

Thank you for this catch we have now add this reference, we agree dengue is an excellent inclusion

(line 64) This context about the number of stars in the observable Universe is fantastic.

Thank you for this enthusiastic comment!

(lines 400-401) The choice of weights used for the multiple loss functions is not clear. It would be good to have a better idea of how they were chosen. This is a serious issue given the authors' recommendation in the discussion (around line 334) that multi-objective fitting should be preferred over purely sequential approaches. That's much easier to do when the weights for the multiple objectives are pre-specified and taken on faith as appropriate.

We agree. In balance of the significant improvement and number of changes to our calibration algorithms we chose to keep the weighting consistent with previous calibrations of OpenMalaria, primarily for comparability. The weights were chosen so that all objectives contribute approximately somewhat equally to $F(\theta)$, however with some higher weighting for severe outcome. This is mentioned in line 455.

(lines 403-404) It would be helpful if more could be said about the minimum number of evaluations needed to inform the emulator function. Convergence is mentioned on line 413, but this is rather vague.

We have replaced the word "converges" with "improves". We decided against providing a hard minimum number of evaluations, and in our set up the end point of our calibration was when no more improvement occurred, or we hit memory limits.

(line 458) It might be useful to add a sentence with a bottom-line statement of the total number of model simulations required for this calibration procedure.

In our case, we were able to complete approximately 130,000 simulations before we hit the memory limit (added to line 190). However, this number will vary depending on

many choices made during implementation, e.g. the number of replicates, size of the initial set, computing system and available memory, and complexity of the solution space, which is simulator-dependent.

(line 188 and on) The results about performance are quite interesting. I would expect faster performance than previous methods, but a better fit overall is a nice cherry on top. Well done.

Thank you!

(line 194) What is a "satisfactory" fit? Something a little bit worse than the fit obtained? I'll have to remember this trick! ;)

This was previously defined in Smith 2012 as the point where the model sufficiently captured the disease biology and was set as a goal end point for calibration in the documentation of OpenMalaria in previous efforts. We decided to keep this definition for comparability to previous calibration efforts.

(line 238) Missing word to begin the paragraph?

Thank you for the catch, something got lost here. We added the rest of the sentence back in.

(discussion) Any more commentary that the authors might be able to provide for how their approach could be applied to other agent-based models would be valuable. Are there unique aspects of OpenMalaria and its well trodden principles for fitting to data that make this approach more successful here than it might be for other models?

The code is available on GitHub and easily adaptable to the calibration of any simulation model. The number of input parameters and objective functions are kept flexible (to allow for adaptation to other model variants of OpenMalaria, thus also adaptable for other models. Loss-functions are particular to the data and model output). To adapt the code to other simulators, researchers would have to a) rewrite the code that runs the model (in our case "EvalOM.R") and tailor the setup to their respective operating system (in our case this was the CentOS on the Basel University computing cluster). Requirements to adapt this workflow are sufficient calibration data, and a per-objective goodness-of-fit. We have added a generalised statement about the adaptability of the workflow to the end of the manuscript but would be happy to provide individual support in the specific tailoring of this workflow to the calibration of other simulators, should this be required.

We added at line 455

The workflow presented here provides great advances in the calibration of detailed mathematical models of infectious diseases such as IBMs. Provided sufficient calibration data to determine goodness-of-fit, our approach is easily adaptable to any agent-based model and could become the new modus operandi for multi-objective, high-dimensional calibration of stochastic disease simulators.

And in the methods line 665

The full algorithm code is available on GitHub (https://github.com/reikth/BayesOpt_Calibration) and can be easily adapted to calibrate any simulation model. The number of input parameters and objective functions are flexible, thus, to adapt the code to other simulators code should be updated to run the model simulator and tailored to user's operating system. Further requirements to adapt the workflow are sufficient calibration data, and a per-objective goodness-of-fit.

Reviewers' Comments:

Reviewer #1:

Remarks to the Author:

I am happy with the authors responses. I still think the marriage of NNs and GPs is awkward. Many strides have been made to expand GP modeling to larger numbers of training examples and input dimension without compromising on uncertainty quantification. Meanwhile, UQ for deep learning remains illusive. Nevertheless, I like that this paper is promoting the use of surrogate modeling and calibration of computer experiments tools for epidemiological analysis.

Reviewer #2:

Remarks to the Author:

I am satisfied with the revisions.